# Effects of *Forsythia Suspense* Extract as an Antibiotics Substitute on Growth Performance, Nutrient Digestibility, Serum Antioxidant Capacity, Fecal *Escherichia coli* Concentration and Intestinal Morphology of Weaned Piglets

**DOI:** 10.3390/ani9100729

**Published:** 2019-09-26

**Authors:** Shenfei Long, Li Liu, Sujie Liu, Shad Mahfuz, Xiangshu Piao

**Affiliations:** State Key laboratory of Animal Nutrition, College of Animal Science and Technology, China Agricultural University, Beijing 100193, China; longshenfei@cau.edu.cn (S.L.); jiahao6468@163.com (L.L.); heiluobo12300@163.com (S.L.); shadmahfuz@yahoo.com (S.M.)

**Keywords:** *Forsythia suspense* extract, performance, serum antioxidant status, *Escherichia coli*

## Abstract

**Simple Summary:**

Weaning stress may reduce feed intake, weight gain and health status of piglets. Antibiotics are used to overcome post-weaning disorders. However, the abuse of antibiotics in pig feed has become a worldwide problem. Previous studies show Chinese herbs have been used as a potential non-antibiotic way to enhance anti-inflammatory and anti-microbial functions of piglets. This study aims to evaluate the effect of *Forsythia suspense* extract (FSE) as an antibiotics substitute on performance, nutrient digestibility, serum antioxidant capacity, fecal *Escherichia coli* concentration and intestinal morphology of weaned piglets. The results show that dietary FSE supplementation can substitute antibiotics in improving antioxidant capacity, nutrients digestibility and reducing fecal *E. coli* content, so as to reduce nitrogen output and diarrhea rate, and eventually enhance growth performance in weaned piglets.

**Abstract:**

The aim of this study is to determine the efficiency of *Forsythia suspense* extract (FSE) as an antibiotics substitute on performance, nutrient digestibility, serum antioxidant capacity, fecal *Escherichia coli* concentration and intestinal morphology of weaned piglets. A total of 108 Duroc × (Landrace × Yorkshire) weaned piglets (28 days (d) weaned, average body weight of 8.68 ± 1.36 kg) were randomly assigned into three dietary treatments, six pens per treatment, three barrows and three gilts per pen. The treatments contained a corn-soybean meal basal diet (CTR), an antibiotic diet (basal diet + 75 mg/kg chlortetracycline; CTC), and an FSE diet (basal diet + 200 mg/kg FSE; FSE). The experiment included phase 1 (d 1 to 14), phase 2 (d 15 to 28) and phase 3 (d 29 to 35). Compared with CTR, piglets fed FSE show improved (*p* < 0.05) average daily gain (ADG) and average daily feed intake in phase 2, as well as enhanced (*p* < 0.05) ADG from day 15 to 35 and day 1 to 28. Piglets supplemented with CTC and FSE showed a reduced (*p* < 0.05) diarrhea rate in phase 1, while piglets fed FSE showed enhanced (*p* < 0.05) apparent total tract digestibility (ATTD) of dry matter, organic matter, crude protein and gross energy, as well as lower (*p* < 0.05) nitrogen output in phase 2 compared with CTR and CTC. The content in the form of Colony-Forming Units (CFUs) of fecal *E. coli* on day 14 and 28 was lower (*p* < 0.05) in piglets fed FSE in comparison with CTR. The contents of total antioxidant capacity, superoxide dismutase and catalase in serum are enhanced (*p* < 0.05) compared with CTR and CTC, whereas the concentration of malondialdehyde in serum was decreased (*p* < 0.05) for piglets fed FSE on day 28 compared with CTC. The villus height to crypt depth ratio in ileum was numerically higher (*p* < 0.05) in piglets fed FSE in comparison with CTR. In conclusion, dietary FSE supplementation could substitute CTC in improving antioxidant capacity, nutrients digestibility and reducing fecal *E. coli* content, so as to reduce nitrogen output and diarrhea rate, and eventually improve performance in weaned piglets.

## 1. Introduction

Weaning is a stressful challenge for piglets because of the sudden changes of physiology and surrounding environment. These weaning stresses, usually presented as oxidative stress [1], may reduce antioxidant status, immunity and intestinal functions, which results in a reduction in feed intake, weight gain and health of piglets, as well as an increase in diarrhea incidence, morbidity and mortality [2]. To overcome these post-weaning disorders caused by oxidative stress, researchers report that the use of antibiotics, such as chlortetracycline (CTC), as in-feed supplements after weaning may help enhance weight gain by 16% and feed utilization by 7%, as well as reduce morbidity and mortality approximately 50% for weaned piglets [3,4]. However, the abuse or misuse of antibiotics in piglets’ feed can result in bacteria resistant to antibiotics and lead to potential residues in animal products (such as pork) and in the environment, which may enhance the possibility of antibiotic-resistant infections in humans [5]. Therefore, the European Union, the United States and many other countries have reduced or prohibited the use of antibiotics in animal feed and found appropriate alternatives for antibiotics.

Studies in our labs have demonstrated that essential oils [6], essential oils combined with mixed organic acids [7], probiotics [8], chito-oligosaccharide [9] as well as natural plant herbs [10] can potentially serve as antibiotic substitutes. Currently, the practice of using traditional herbal medicine is gaining more attention worldwide in animal health care systems [11]. The addition of traditional Chinese herbs, especially *Coptis chinensis* and *Forsythia suspense*, have been used a potential non-antibiotic way to enhance anti-inflammatory and anti-microbial function, antioxidant status and performance in livestock [10,12,13].

The fruit or leaf extract of *Forsythia suspensa* Vahl (Oleaceae) could be widely used as Chinese medicine to help treat some infections, including pharyngitis, nephritis, febrile erysipelas, ulcers, tonsillitis, gonorrhea and acute respiratory syndrome [14]. Previous studies in our lab have demonstrated that lignan, phenethyl alcohol glycoside, volatile oil as well as pentacyclic triterpenoids are the major active compounds of *Forsythia suspensa* extract (FSE) [15,16], which has proven to have anti-oxidant [17], anti-bacterial [12,18], anti-inflammatory [19] and anti-allergy [20] effects. Currently, our lab has used different stress models and different animal categories (e.g., piglets, broilers, laying hens and mice) to demonstrate the antioxidant properties and immune enhancement functions of FSE. Studies also show FSE can enhance performance by modulating intestinal permeability, antioxidant status and immune function in animals combined with chito-oligosaccharide [13,19] or berberine [12]. However, there are few data to clarify the possibility of FSE as an antibiotic substitute and the possible mechanism functions of FSE. Therefore, the objective of this study is to evaluate the efficiency of FSE as an antibiotics substitute on performance, serum antioxidant status, fecal *Escherichia coli* content and intestinal morphology in weaned piglets.

## 2. Materials and Methods

The study was carried out at the Animal Experimental Base (Fengning, Hebei, China) in the National Feed Engineering Technology Research Center of the Ministry of Agriculture Feed Industry Center. All the experimental procedures and operations used in the management and care of piglets were in agreement with the China Agricultural University Laboratory Animals Welfare and Animal Experimental Ethical Inspection (Beijing, China; No. AW09089102-1).

### 2.1. Experimental Products

Dried and ground forsythia fruits (about 100 g) were prepared and extracted by 80% ethanol (500 mL), sonicated for 1 h and then filtered. Ethanol was used to extract the residue twice and then rotary vaporization (Buchi, Rotavapor R-124, Flawil, Switzerland) was utilized to dry and combine the filtrates. The main functional ingredients in FSE are forsythoside A (1.65%), phillyrin (8.17%), forythialan A (4.13%) and phillygenin (1.67%) [10]. An additional antibiotic mixture (chlortetracycline, CTC) was produced by Beijing Tongli Xing Department of Agricultural Science and Technology Company Limited (Beijing, China).

### 2.2. Experimental Animals, Management and Design

A total of 108 Duroc × (Landrace × Yorkshire) weaned piglets (28 days (d) weaned, average body weight of 8.68 ± 1.36 kg) were randomly assigned into 3 dietary treatments, 6 pens per treatment, 3 barrows and 3 gilts per pen. The treatments contained a corn-soybean meal basal diet (CTR), an antibiotic diet (basal diet + 75 mg/kg chlortetracycline; CTC), and an FSE diet (basal diet + 200 mg/kg FSE; FSE). The experiment included phase 1 (d 1 to 14), phase 2 (d 15 to 28) and phase 3 (d 29 to 35). As shown in Table 1, nutrients in the diet met the recommended requirements (National Research Council, NRC, 2012) [21].

All the piglets were raised in experimental pens (1.2 m × 2 m) fitted with a duckbill drinker, an adjustable stainless steel feeder and plastic slatted floors. The piglets were given access to water and fed ad libitum in powder form. Inside the pen, these piglets also had free access to feed (in mash form) and water ad libitum. The environment in the pig house, including the contents of carbon dioxide and ammonium in the air, ventilation intensity, humidity and temperature, was controlled automatically. The average temperature in house was controlled at 24–26 °C, while the relative humidity was maintained at 60%–70%. In order to prevent disease, the experimental house was cleaned every day and the immunization procedure was conducted every week. After 12 h of starvation, the individual weight of piglets and the feed weight of each pen were weighed on day 0, 14, 28 and 35 to calculate the average daily gain (ADG), average daily feed intake (ADFI) and feed efficiency (FE, ADFI/ADG). From day 0 to 28, the diarrhea score was monitored according to the previously described system [22] from 1 to 5: 1, hard firm feces (rarely seen); 2, slightly soft feces; 3, soft, partially formed feces; 4, loose, semi-liquid feces (diarrhea); 5, watery, mucus-like feces (severe diarrhea). The determination of diarrhea rate was mainly dependent on the average diarrhea score following the formula: diarrhea rate (%) = diarrhea days × the number of diarrhea pigs/(experiment days × the total number of pigs) [23].

### 2.3. Experimental Sample Collection and Analysis

During this experiment, a total of 2 kg representative feed samples were collected weekly. From day 12 to 14 and day 26 to 28, the rectal palpation was used to make sure approximately 100 g of fresh feces were collected using the grab sample technique. All the fecal samples were frozen at −20 °C immediately after collection until analysis. The feces collected after 3 days were pooled by pen and dried at 65 °C for 72 h. Before analysis, all these dried feces and feed samples were ground to pass through a 1-mm sieve.

The dry matter (DM), ether extract (EE) and crude protein (CP) of the feed and fecal samples were measured using the methods of Association of Official Agricultural Chemists (AOAC) [24]. The gross energy (GE) in feed and fecal samples was determined by an automatic isoperibolic oxygen bomb calorimeter (Parr 1281, Automatic Energy Analyzer; Moline, IL, USA). Moreover, an atomic absorption spectrophotometer (Z-5000; Hitachi, Tokyo, Japan) was used to determine the content of chromium in feed and fecal samples. Organic matter (OM) was calculated as 1 − ash content (DM base). The calculation formula for total carbohydrates was as follows: the calculation = dry matter − crude protein − ether extract − ash [25]. Nutrient digestibility was determined by the equation as follows: Apparent total tract digestibility_nutrient_ (ATTD) = 1 − (Cr_diet_ × nutrient_feces_)/(Cr_feces_ × nutrient_diet_). The manure nitrogen output from piglets fed FSE during a 28-d period of the experiment was calculated by the equation as follows: fecal nitrogen (N) excretion per weight gain (g/kg) = (N intake (g/d) × (100 − ATTD of N))/([100 × ADG (kg/d)).

For the determination of fecal microbiota, the fresh fecal samples (about 100 g) of piglets in the CTR and FSE groups collected on day 14 and day 28 were first thawed at room temperature. A total of 1 g of fresh fecal sample was taken and transferred into a 9-mL diluent tube and then diluted 6 times serially in order to make sure every sample was fully dissolved. The test of fecal *E. coli* concentrations in fresh fecal samples was carried out within a day after collection. The total content of *E. coli* was determined using Maconkey agar to plate 0.1 mL diluent. An electro-heating standing-temperature cultivator (37 °C) was used to incubate all the petri dishes for 24 h. Before statistical analysis, the fecal *E. coli* concentrations were transformed (Log).

After starvation for 12 h, approximately 8 mL of blood was collected from a piglet near the average group body weight in each pen via jugular vein puncture into a vacutainer at 7:00 a.m. on day 28 (Becton Dickinson Vacutainer Systems, Franklin Lakes, NJ, USA). After stewing for 3 h, all the collected blood samples were centrifuged at 3000× *g* for 15 min at 4 °C to get the serum samples, which were also stored at −20 °C until analysis. Estimation of triglyceride (TG) and blood urea nitrogen (BUN) contents in serum were conducted by Hitachi 7600 Automatic Biochemical Instrument. An ELISA kit (IgG, IgM and IgA quantitation kit; Bethyl Laboratories, Inc., Montgomery, TX, USA) was used to determine the concentrations of serum immunoglobulins (including immunoglobulin G, immunoglobulin M and immunoglobulin A). Determination of total antioxidant capacity (T-AOC), catalase (CAT), superoxide dismutase (SOD), glutathione peroxidase (GSH-Px) and malondialdehyde (MDA) levels in serum were conducted by spectrophotometric methods using a spectrophotometer (Leng Guang SFZ1606017568, Shanghai, China) following the instructions of the kit’s manufacturer (Nanjing Jiancheng Bioengineering Institute, Nanjing, China).

On day 28, the aseptic duodenal, jejunal and ileal samples (about 5 cm fragment in the middle of each intestine, duodenum was selected as the proximal 1/3 of the small intestine, jejunum as the 1/3 mid and ileum as 1/3 distal part) were collected from slaughtered barrows (near the average group body weight) selected in each pen for the determination of intestinal morphology. Then, 10% neutral buffered formalin was used to fix rapidly these histological samples for slicing. After 48 h of fixation, the sections of intestinal tissues were washed, excised, dehydrated, as well as embedded in the paraffin wax, and then 5 transverse sections were sliced, installed on glass slides and dyed with eosin and hematoxylin. At least 20 orientated villi and their adjoining crypts were selected randomly on each slice and measured to calculate the average villus height and crypt depth via a light microscope in small caps using a calibrated 10-fold eyepiece graticule. The ratio of villus height to crypt depth was calculated and used for further analysis.

### 2.4. Statistical Analysis

The mixed model of SAS (version 9.2, 2008) [26] was used for variance analysis of all the data. The dietary treatments were fixed effects, while sex and body weight of pigs were the random effects. For the analysis of growth performance and diarrhea rate, the pen was treated as the statistical unit, whereas for the analysis of other data, individual piglets were taken as a statistical unit. The Student–Newman–Keul multiple range test was used for determining the statistical differences among all the treatments. Significant difference between the mean value was defined at *p* ≤ 0.05, while a trend for the significance between the mean value was designated at 0.05 < *p* ≤ 0.10.

## 3. Results

### 3.1. Performance and Diarrhea Rate of Piglets

The growth performance (ADG, ADFI and FE) and diarrhea rate are shown in Table 2. Compared with CTR, the ADG and ADFI are improved (*p* < 0.05) approximately 19% and 17% in piglets supplemented with an FSE diet from day 15 to 28, and the ADG is enhanced (*p* < 0.05) about 12% and 13% by FSE for piglets from day 1 to 28 and day 15 to 35. Meanwhile, piglets fed an FSE diet also showed a tendency in increasing ADG from day 1 to 35 in comparison with CTR (*p* = 0.10). The FE was not affected by any dietary additives. Compared with CTR, piglets fed CTC and FSE had a reduced (*p* < 0.05) diarrhea rate of about 77% and 61% from day 1 to 14.

### 3.2. The ATTD of Nutrients and Nitrogen Output

The ATTD of nutrients is presented in Table 3. In phase 1, piglets fed FSE showed increased (*p* < 0.05) ATTD of DM, OM, CP and GE by approximately 7%, 6%, 9% and 7%, respectively compared with CTR and CTC. In phase 2, there was a tendency of improvement on the ATTD of CP (*p* = 0.06) and GE (*p* = 0.10) in piglets supplemented with FSE in comparison with CTR and CTC.

The manure nitrogen output from piglets fed FSE during a 28-d period experiment is presented in Figure 1; piglets fed FSE have lower (*p* < 0.05) nitrogen output compared with CTR and CTC.

### 3.3. Fecal Escherichia coli Contents

The fecal *E. coli* content on day 14 and 28 is given in Table 4. The content [in the form of the Colony-Forming Units (CFUs)] of *E. coli* in feces is lower (*p* < 0.05) by approximately 59% and 36% on day 14 and 28, respectively in piglets fed FSE in comparison with CTR.

### 3.4. Serum Metabolic Profile, Immunity and Antioxidant Indices

Effects of FSE on serum metabolic profile, immunity and antioxidant status of weaned piglets on day 28 is shown in Table 5, Table 6 and Table 7. Compared with CTR, there is no significant difference of serum metabolic profile and immunoglobulin levels in piglets fed diets supplemented with FSE. However, the serum contents of T-AOC, SOD and CAT are enhanced (*p* < 0.05) in piglets fed FSE compared with those fed CTR and CTC, whereas the concentration of MDA is decreased (*p* < 0.05) in piglets fed FSE on day 28 in comparison with CTC.

### 3.5. Intestinal Morphology

The effects of FSE on intestinal morphology of weaned piglets on day 28 are shown in Table 8. There is a decreased tendency of crypt depth in ileum of piglets fed FSE (*p* = 0.09), while the villus height to crypt depth ratio is numerically higher (*p* < 0.05) in piglets fed FSE in comparison with CTR. The villus height of piglets fed FSE are numerically higher (*p* > 0.05) at approximately 11% in duodenum and 26% in jejunum, respectively.

## 4. Discussion

Weaning stress may cause a reduction of growth rate and feed intake in piglets, while current results show dietary FSE supplementation can enhance ADG and ADFI in phase 2, but there is no difference of performance among dietary treatments in phase 1. Similar results are also shown in the study by Han et al. [18], which demonstrated that during the first period, performance has no difference in the FSE group, while it experiences a significant enhancement during the finisher and overall phase in broilers. This indicates that the beneficial effect of FSE on piglets may relate to its cumulative effect in animals. However, according to the study of Zhao et al. [19], dietary supplementation with 100 mg/kg FSE can improve ADG and FE in the first 2 weeks. These different findings may relate to the amount and composition of FSE in different studies. The present study showed that there is no significant difference of FE between treatments, while other studies show dietary plant polyphenols supplementation can improve the FE in pigs via the improvement of nutrients digestibility [27] and the enhancement of health status via inhibiting inflammation [28]. These contradictory functions of FSE on FE in different studies may also relate to the type, additive amount and chemical composition of Chinese plant polyphenols. The current study also shows that the effects of FSE on enhancing growth performance is better than CTC, which is partly in line with the study of Han et al. [18], who also indicated that FSE has the potential to replace antibiotics in improving the performance and intestinal health of animals. The reason for the improved performance may also be due to the decreased diarrhea rate and improved nutrient utilization of piglets fed an FSE diet.

After weaning, piglets are susceptible to disease due to environmental changes, which usually cause severe post-weaning diarrhea. The present study shows that piglets fed a CTC and FSE diet experienced a significant reduction in diarrhea rates in phase 1 and 2; this could possibly be due to the lower content of *E. coli* in feces of piglets fed a CTC and FSE diet since diarrhea caused by *E. coli* is a major challenge for post-weaning pigs [8]. The present result is in agreement with the study of Han et al. [18], who reported that FSE can regulate intestinal flora via reducing the cecal *E. coli* counts in vivo and inhibiting the reproduction of *E. coli* K88, *Staphylococcus aureus* and *salmonella* in vitro. The result may be because polyphenol in FSE can increase the fecal pH value and lower the concentrations of volatile fatty acids [27]. Another reason may be that the forsythiaside and phillyrin in FSE have strong broad-spectrum antimicrobial activity [29], which can effectively inhibit *E. coli*, *Pseudomonas aeruginosa* and *S. aureus* [30,31]. Moreover, the essential oils from FSE have also demonstrated their effectiveness in inhibiting the growth of *S. aureus*, *Bacillus subtilis*, *E. coli*, *P. aeruginosa*, *Candida albicans* and *Aspergillus Niger* [32]. In addition, according to a study by Zhang et al. [12], FSE may also increase the level of *Lactobacillus* while reducing the level of *E. coli* in the cecum of broilers on day 21 and 42, thus improving the structure of intestinal flora in broilers. In our present study, we consider the number of CFUs of *E. coli* only; the differences may not represent a real reduction in *E. coli*, therefore, this finding still needs to be further estimated in our following studies.

During the first two weeks after weaning, piglets may face severe challenges when they utilize nutrients, which is mainly due to disorders in the digestive and absorption systems. In the present study, the ATTD of nutrients were enhanced in piglets fed an FSE diet in phase 1, which is partly in agreement with the previous study of Han et al. [18] and Zhang et al. [12]. The digestibility of nutrients strongly increased in piglets fed FSE compared to CTR in phase 1 and tended to be enhanced in phase 2, which can normally lead to an improvement of ADG and FE [19]. However, our study only found the enhancement of ADG in phase 2, from day 15 to 35 and day 1 to 28, which may be due to that the enhancement of ADFI in phase 2 as well as the difference of the additive amount and composition of FSE. After weaning within 2 weeks, the digestive system of piglets is not well developed and Chinese medicine usually has an accumulative effect for animals, which may lead to the positive effect of FSE on ADG to be easier to find in phase 2, from day 15 to 35 and day 1 to 28 [18]. Yet this finding still remains to be investigated in further study. Moreover, the improved nutrient digestibility in the current study is in line with the increased villus height to crypt depth ratio and decreased crypt depth in ileum of piglets fed an FSE diet. This may be because FSE can help improve small intestinal villus morphology, enhancing the growth of villus as well as the villus height to crypt depth ratio, which in turn may increase the absorption of DM and CP [12]. In addition, FSE can also improve the proliferation of peripheral blood lymphocytes and intestinal permeability, so as to increase the intestinal capacity to utilize nutrients [20]. The higher numerical changes of the villus height, crypt depth and their ratio in the small intestine probably indicates FSE may benefit and help in the development of the small intestine in weaned piglets [18]. A possible reason for the obviously improved nutrient digestibility in the early phase for piglets may be that after weaning, the development of the intestine and activity of digestible enzymes are poor, while the intestine is better developed after two weeks in the later phase. The reduction of manure N output from piglets fed diets based on a corn-soybean meal diet supplemented with FSE is mainly due to the improved ATTD of CP.

When oxidants and antioxidants are in an unbalanced state (oxidative stress reaction), a large number of reactive oxygen species (ROS) will be produced. These excessive ROS and free radicals can lead to lipid peroxidation, DNA damage, cell apoptosis or cell cycle arrest, eventually leading to cell death [33]. In the present study, the contents of T-AOC, SOD and CAT are enhanced, whereas the concentration of MDA is decreased in piglets fed an FSE diet. This may because FSE could be an effective antioxidant in vitro and in vivo. In vitro studies have shown that the scavenging rates of 100 g/mL and 250 g/mL FSE for 1,1-diphenyl-2-trinitrophenylhydrazine (DPPH) free radical can reach 77.2% and 81.3% respectively, which are comparable to vitamin C [17]. Moreover, the phenolic hydroxyl ortho-position on the phenyl ring of forsythiaside in FSE has a strong antioxidant and free radical scavenging ability, which can effectively protect cells from hydrogen peroxide-induced cell damage (reducing ROS and MDA levels) and mitochondrial-dependent cell apoptosis [10]. In vivo studies have shown that FSE can effectively alleviate the oxidative damage of diquat to Sprague-Dawley (SD) rats, reduce MDA content and inflammatory factors in the liver and serum [10], which is mainly because forsythiaside in FSE can activate non-enzymatic system, improve total antioxidant capacity and increase the expression of antioxidant enzymes in cells by promoting the level of nuclear factor erythroid-2-related factor 2 in the nucleus [29], so as to activate antioxidant enzymes system (such as SOD, CAT and GSH-Px). Moreover, the polyphenols in FSE can also play an important role in enhancing the activities of antioxidant enzymes in the blood, liver, spleen and kidneys, which largely contributes to improving the health status and growth performance of weaned piglets [34].

## 5. Conclusions

In conclusion, dietary *Forsythia suspense* extract (200 mg/kg) supplementation as a substitute for chlortetracycline can help improve performance, nutrient digestibility, serum antioxidant capacity and intestinal morphology as well as reduce the content of *Escherichia coli* in the feces of weaned pigs.

## Figures and Tables

**Figure 1 animals-09-00729-f001:**
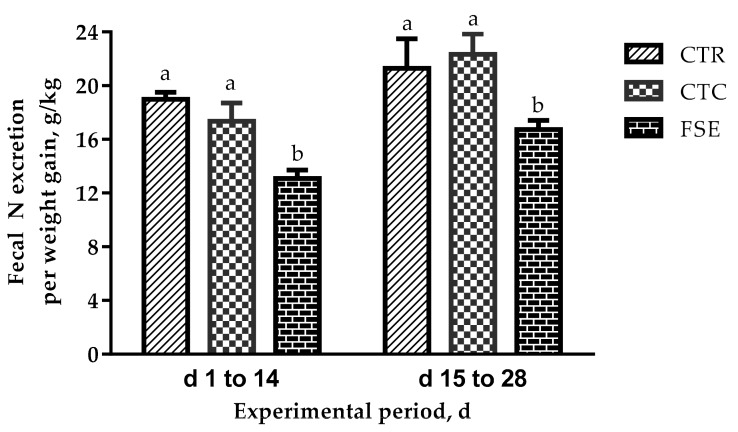
Manure nitrogen (N) output from piglets fed diets based on corn soybean meal diet supplemented with 75 mg/kg chlortetracycline or 200 mg/kg *Forsythia suspense* extract during a 28-d period experiment. Values are least square means ± SEM, *n* = 6/treatment. ^a,b^ Means without common letters differ significantly (*p* < 0.05).

**Table 1 animals-09-00729-t001:** Composition and nutrient levels of basal diets (%, as-fed basis).

Ingredients	Day 1 to 14	Day 15 to 35
Corn	55.64	56.00
Soybean meal, 43%	18.00	20.00
Extruded soybean	12.00	10.00
Spray dried plasma protein	4.00	0.00
Fish meal	2.00	4.00
Whey powder	2.00	5.50
Soy oil	2.80	1.73
Dicalcium phosphate	0.90	0.52
Limestone	1.12	0.80
Salt	0.30	0.30
L-lysine HCl, 78%	0.22	0.26
DL-Methionine, 98%	0.10	0.06
L-Threonine, 98%	0.01	0.07
L-Tryptophan, 98%	0.00	0.01
Zinc oxide	0.16	0.00
Chromic oxide	0.25	0.25
Vitamin-mineral premix ^1^	0.50	0.50
Calculated nutrient levels		
Digestible energy, kcal/kg	3541	3490
Crude protein	21.47	20.05
Calcium	0.80	0.70
Digestible phosphorus	0.40	0.33
Standardized ileal digestible lysine	1.35	1.23
Standardized ileal digestible methionine	0.39	0.36
Standardized ileal digestible threonine	0.79	0.73
Standardized ileal digestible tryptophan	0.23	0.20
Analyzed nutrient levels		
Gross energy, kcal/kg	4016	3996
Crude protein	21.56	19.78
Calcium	0.82	0.72
Gross phosphorus	0.60	0.55
Lysine	1.52	1.42
Methionine	0.45	0.42
Threonine	0.90	0.87
Tryptophan	0.27	0.25

^1^ premix for each kg diet: vitamin A, 12,000 IU; vitamin D_3_, 2500 IU; vitamin E, 30 IU; vitamin K_3_, 30 mg; vitamin B_12_, 12 μg; riboflavin, 4 mg; pantothenic acid, 15 mg; nicotinic acid, 40 mg; choline chloride, 400 mg; folic acid, 0.7 mg; vitamin B_1_, 1.5 mg; vitamin B_6_, 3 mg; biotin, 0.1 mg; manganese, 40 mg; iron, 90 mg; zinc, 100 mg Mg; copper, 8.8 mg; iodine, 0.35 mg; selenium, 0.3 mg.

**Table 2 animals-09-00729-t002:** Effects of *Forsythia suspense* extract on performance and diarrhea rate in weaned piglets.

Item	CTR ^1^	CTC ^1^	FSE ^1^	SEM	*p*-Value
day 1 body weight	8.68	8.68	8.68	0.01	0.85
day 1 to 14					
Average daily gain, g	404	418	422	14.84	0.68
Average daily feed intake, g	748	783	773	39.57	0.82
Feed efficiency, g gain/g feed	0.55	0.54	0.55	0.03	0.97
Diarrhea rate, %	3.09 ^a^	0.71 ^b^	1.19 ^b^	0.44	0.01
day 15 to 28					
Average daily gain, g	453 ^b^	472 ^b^	540 ^a^	14.11	<0.01
Average daily feed intake, g	903 ^b^	956 ^a,b^	1060 ^a^	37.00	0.04
Feed efficiency, g gain/g feed	0.50	0.49	0.51	0.02	0.79
Diarrhea rate, %	4.05	4.28	3.57	1.32	0.93
day 1 to 28					
Average daily gain, g	428 ^b^	445 ^b^	481 ^a^	7.80	<0.01
Average daily feed intake, g	825	870	916	30.13	0.16
Feed efficiency, g gain/g feed	0.52	0.51	0.53	0.02	0.82
Diarrhea rate, %	3.57	2.50	2.38	0.74	0.49
day 15 to 35					
Average daily gain, g	561 ^b^	587 ^b^	633 ^a^	16.41	0.04
Average daily feed intake, g	1005	1031	1075	24.89	0.20
Feed efficiency, g gain/g feed	0.56	0.57	0.59	0.02	0.56
day 29 to 35					
Average daily gain, g	670	702	726	30.29	0.45
Average daily feed intake, g	1107	1106	1090	37.96	0.94
Feed efficiency, g gain/g feed	0.60	0.63	0.67	0.02	0.23
day 1 to 35					
Average daily gain, g	549	574	603	17.40	0.10
Average daily feed intake, g	966	988	1003	24.42	0.58
Feed efficiency, g gain/g feed	0.57	0.58	0.60	0.02	0.37

SEM means standard error of mean. ^a,b^ Different superscripts within a row indicate a significant difference (*p* < 0.05). ^1^ CTR: control; CTC: chlortetracycline; FSE: *Forsythia suspense* extract.

**Table 3 animals-09-00729-t003:** Effects of *Forsythia suspense* extract on apparent total tract digestibility of nutrients in weaned piglets (%).

Items	CTR ^1^	CTC ^1^	FSE ^1^	SEM	*p*-Value
day 14					
Dry matter	81.63 ^b^	82.92 ^b^	87.31 ^a^	0.91	<0.01
Organic matter	83.93 ^b^	85.08 ^b^	88.72 ^a^	0.81	<0.01
Crude protein	75.47 ^b^	76.94 ^b^	82.51 ^a^	1.34	0.01
Gross energy	81.16 ^b^	82.65 ^b^	87.07 ^a^	0.92	<0.01
Total carbohydrates	89.05 ^b^	89.96 ^b^	92.37 ^a^	0.56	<0.01
day 28					
Dry matter	77.21	74.83	78.10	1.03	0.12
Organic matter	80.02	78.10	81.01	0.92	0.13
Crude protein	66.43	62.83	68.81	1.48	0.06
Gross energy	77.06	74.77	78.95	1.18	0.10
Total carbohydrates	86.08	85.05	86.85	0.71	0.25

SEM means standard error of mean. ^a,b^ Different superscripts within a row indicate a significant difference (*p* < 0.05). ^1^ CTR: control; CTC: chlortetracycline; FSE: *Forsythia suspense* extract.

**Table 4 animals-09-00729-t004:** Effects of *Forsythia suspense* extract on *Escherichia coli* content in feces of weaned piglets (10^8^ CFUs)/g).

Items	CTR ^1^	FSE ^1^	SEM	*p*-Value
day 14				
*Escherichia coli*	28.75 ^a^	14.08 ^b^	1.55	0.02
day 28				
*Escherichia coli*	21.52 ^a^	13.88 ^b^	0.83	0.02

SEM means standard error of mean. CFUs means Colony-Forming Units. ^a,b^ Different superscripts within a row indicate a significant difference (*p* < 0.05). ^1^ CTR: control; FSE: *Forsythia suspense* extract.

**Table 5 animals-09-00729-t005:** Effects of *Forsythia suspense* extract on serum metabolic profile of weaned piglets on day 28 (mmol/L).

Items	CTR ^1^	CTC ^1^	FSE ^1^	SEM	*p*-Value
Triglyceride	1.79	1.99	1.99	0.08	0.20
Blood urea nitrogen	2.36	3.11	2.51	0.61	0.67

SEM means standard error of mean. ^1^ CTR: control; CTC: chlortetracycline; FSE: *Forsythia suspense* extract.

**Table 6 animals-09-00729-t006:** Effects of *Forsythia suspense* extract on serum immunoglobulin levels of weaned piglets on day 28 (g/L).

Items	CTR ^1^	CTC ^1^	FSE ^1^	SEM	*p*-Value
Immunoglobulin A	0.97	0.94	1.09	0.09	0.50
Immunoglobulin G	21.16	20.85	21.23	0.51	0.85
Immunoglobulin M	2.33	2.37	2.40	0.04	0.47

SEM means standard error of mean. ^1^ CTR: control; CTC: chlortetracycline; FSE: *Forsythia suspense* extract.

**Table 7 animals-09-00729-t007:** Effects of *Forsythia suspense* extract on antioxidant status in serum of weaned piglets on day 28.

Items	CTR ^1^	CTC ^1^	FSE ^1^	SEM	*p*-Value
T-AOC ^2^, U/mL	13.53 ^b^	10.20 ^c^	14.43 ^a^	0.25	<0.01
SOD ^2^, U/mL	73.04 ^b^	64.79 ^c^	82.36 ^a^	1.88	<0.01
CAT ^2^, U/mL	44.49 ^b^	36.61 ^c^	63.24 ^a^	2.06	<0.01
GSH-Px ^2^, U/mL	809	796	843	14.86	0.15
MDA ^2^, nmol/mL	4.91 ^b^	5.60 ^a^	4.41 ^b^	0.56	<0.01

SEM means standard error of mean. ^a,b^ Different superscripts within a row indicate a significant difference (*p* < 0.05). ^1^ CTR: control; CTC: chlortetracycline; FSE: *Forsythia suspense* extract. ^2^ T-AOC: total antioxidant capacity; SOD: superoxide dismutase; CAT: catalase; GSH-Px: glutathione peroxidase; MDA: malondialdehyde.

**Table 8 animals-09-00729-t008:** Effects of *Forsythia suspense* extract on intestinal morphology of weaned piglets on day 28.

Items	CTR ^1^	CTC ^1^	FSE ^1^	SEM	*p*-Value
Duodenum					
Villus height	468	561	519	53.40	0.53
Crypt depth	314	264	274	64.70	0.85
Villus height/crypt depth	1.55	2.55	2.08	0.52	0.47
Jejunum					
Villus height	419	496	526	46.41	0.34
Crypt depth	269	280	237	39.94	0.74
Villus height/crypt depth	1.60	1.81	2.36	0.23	0.17
Ileum					
Villus height	488	453	440	43.18	0.74
Crypt depth	304	216	180	29.73	0.09
Villus height/crypt depth	1.64 ^b^	2.14 ^a,b^	2.52 ^a^	0.10	<0.01

SEM means standard error of the mean. ^a,b^ Different superscripts within a row indicate a significant difference (*p* < 0.05). ^1^ CTR: control; CTC: chlortetracycline; FSE: *Forsythia suspense* extract.

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
