# Peer review of "Effects of Forsythia Suspense Extract as an Antibiotics Substitute on Growth Performance, Nutrient Digestibility, Serum Antioxidant Capacity, Fecal Escherichia coli Concentration and Intestinal Morphology of Weaned Piglets"

_animals, 2019, doi:10.3390/ani9100729_

Round 1

Reviewer 1 Report

This manuscript reports a study dealing with the effects of a Forsythia Suspense Extract (FSE) on growth performance and several parameters in in weaned piglets. The study is technical sound and the subject is of interest for scientists in the related field of research. The manuscript is well written and easy to read and understand. There are however some points to be addressed:

A chemical characterisation of FSE would be helpful for interpretation of this study. It would be helpful to know the major potential biological active compounds. Lines 95/96 – Table 1: There is a inconsistency between text and Table 1 in the number of phases (text: 3 phases, Table: 2 phases). Please correct! Line 177: ADG was not different in d 29 top 35. Please correct! Table 2: Please add unit of feed efficiency (g gain/g feed?) Table 3: Data in d 14: It is shown that digestibility of nutrients is strongly increased in FSE compared to CTR. These should normally lead to an improvement of feed efficiency, which however was not occurring (Table 2). Please discuss these contradictory results. Figure 1: Y-axis must start at 0 g/kg. Table 5: These data do not have a relevance in piglets, and therefore could be removed (Total cholesterol, LDL, HDL are relevant for the development of atherosclerosis in humans, but do not have any meaning in pigs) Discussion: There are already several studies dealing with plant extracts in piglets. In the Discussion, mainly Chinese studies are cited while other studies were ignored. The authors should also refer to studies coming from other countries. For instance, the studies of Gessner et al. and Fiesel et al. showed results of plant extracts which were very similar to those obtained in the present study. Moreover, results have been reviewed in various publications (e.g. Gessner et al., Tufarelli et al.). These publications should be considered in the Discussion of the results.

Author Response

Revision Note, (List of modification) Date: 2019-09-18 (y-m-d)

Manuscript ID: animals-595611-RI.

Dear Sir

Good day. Thank you very much for your kind consideration with our submitted article and offering us the further opportunity to submit the revised manuscript. Please find here the point to point expert reviewer’s and editor’s comments with necessary changes as per suggested with this attached file, and the amendments are highlighted in red in the revised manuscript. We have revised our manuscript for language and grammar checked by a native English speaker working in our University. We do thanks to skilled reviewers, academic editors and editorial board members as well for their critical evaluation to make the manuscript more effective for review process in Animals Journal.

Many thanks.

Sincerely yours,

Prof. Dr. Xiang Shu Piao,

State Key laboratory of Animal Nutrition, College of Animal Science and Technology, China Agricultural University, Beijing 100193, China

Corresponding Author,

Email: piaoxsh@cau.edu.cn; Tel.:+86-1062733588/Fax.: +86-1062733688

Comments: (Reviewer-1)

Total Comments: A chemical characterisation of FSE would be helpful for interpretation of this study. It would be helpful to know the major potential biological active compounds. Lines 95/96 – Table 1: There is a inconsistency between text and Table 1 in the number of phases (text: 3 phases, Table: 2 phases). Please correct! Line 177: ADG was not different in d 29 top 35. Please correct! Table 2: Please add unit of feed efficiency (g gain/g feed?) Table 3: Data in d 14: It is shown that digestibility of nutrients is strongly increased in FSE compared to CTR. These should normally lead to an improvement of feed efficiency, which however was not occurring (Table 2). Please discuss these contradictory results. Figure 1: Y-axis must start at 0 g/kg. Table 5: These data do not have a relevance in piglets, and therefore could be removed (Total cholesterol, LDL, HDL are relevant for the development of atherosclerosis in humans, but do not have any meaning in pigs) Discussion: There are already several studies dealing with plant extracts in piglets. In the Discussion, mainly Chinese studies are cited while other studies were ignored. The authors should also refer to studies coming from other countries. For instance, the studies of Gessner et al. and Fiesel et al. showed results of plant extracts which were very similar to those obtained in the present study. Moreover, results have been reviewed in various publications (e.g. Gessner et al., Tufarelli et al.). These publications should be considered in the Discussion of the results.

Response: Thank you very much for your critical evaluation and suggestions. We have followed all of your advises in this revised submission.

The details are as follows:

Comments: A chemical characterisation of FSE would be helpful for interpretation of this study. It would be helpful to know the major potential biological active compounds.

Response: Thank you very much, we have added the major potential biological active compounds in materials and methods, the main composition of FSE are forsythoside A, phillyrin, forythialan A, and phillygenin. The levels of them were 1.65%, 8.17%, 4.13% and 1.67%.

Comments: Lines 95/96 – Table 1: There is an inconsistency between text and Table 1 in the number of phases (text: 3 phases, Table: 2 phases). Please correct!

Response: Thank you very much, I have corrected the mistakes in Table 1, I have used the d 1 to 14 and d 15 to 35 to replace the previous one.

Comments: Line 177: ADG was not different in d 29 to 35. Please correct!

Response: Thanks for your reminding, I have corrected the mistake and changed the sentence into the ADG is enhanced (p < 0.05) about 12% and 13% by FSE for piglets from d 1 to 28 and d 15 to 35.

Comments: Table 2: Please add unit of feed efficiency (g gain/g feed?)

Response: Thanks for your reminding, I have also added the unit of feed efficiency.

Comments: Table 3: Data in d 14: It is shown that digestibility of nutrients is strongly increased in FSE compared to CTR. These should normally lead to an improvement of feed efficiency, which however was not occurring (Table 2). Please discuss these contradictory results.

Response: Thanks for your reminding. We have discussed some possible reasons of these contradictory results in discussion. These contradictory function may be due to that the enhancement of ADFI in phase 2 as well as the difference of additive amount and composition of FSE. After weaning within 2 weeks, the digestive system of piglets is not well developed and the Chinese medicine usually has the accumulative effect for animals, which may lead to the positive effect of FSE on ADG easier to be found in phase 2, from d 15 to 35 and d 1 to 28. But this finding still remains to be investigated in further study.

Comments: Figure 1: Y-axis must start at 0 g/kg.

Response: Thanks for your reminding. I have corrected the mistakes in figure 1.

Comments: Table 5: These data do not have a relevance in piglets, and therefore could be removed (Total cholesterol, LDL, HDL are relevant for the development of atherosclerosis in humans, but do not have any meaning in pigs)

Response: Thanks for your reminding, I have deleted these data.

Comments: Discussion: There are already several studies dealing with plant extracts in piglets. In the Discussion, mainly Chinese studies are cited while other studies were ignored. The authors should also refer to studies coming from other countries. For instance, the studies of Gessner et al. and Fiesel et al. showed results of plant extracts which were very similar to those obtained in the present study. Moreover, results have been reviewed in various publications (e.g. Gessner et al., Tufarelli et al.). These publications should be considered in the Discussion of the results.

Response: Thanks for your reminding, I have considered these ideas in the discussion part and added some publications (e.g. Gessner et al., Fiesel et al., Dhama et al.) in the Introduction and Discussion part of our study.

Thank you

Reviewer 2 Report

The authors present a report on the effects of FSE on performance of weaned pigs, with additional evaluation of intestinal morphology, oxidative and immune status. Overall, there is an effect of FSE on anti-oxidant capacity while other parameters measured show minimal or no changes.

The study is interesting in terms of providing further evidence of the potential use of FSE in animal production. In particular, the data on nutrient digestibility is very interesting but not clearly supported by any other changes. However, there are issues with the document that need to be clarified and resolved:

The document needs language and grammar revision.

Write in past tense when referring to experimental procedures and in present when referring to presenting results.

There is no statement on intestinal morphology in the abstract, although it is in the title. Same applies for fecal microbiota. 

Lines 54-55. What is the evidence of residual antibiotics in pork? I believe the concerns about use of antibiotics as growth promoters is not properly addressed here.

The pigs used in this study were weaned at 28 days, which is a bit late considering common practices and can affect the incidence of diarrhea and weight loss in weaned pigs. What was the rationale behind this decision?

Line 111: 12 h fasting.

Why were the animals fasted before weighing?

The terms microbiota is not used properly in the manuscript. Microbiota refers to the microbial communities found in the gastrointestinal tract. In this study, only E. coli was detected. The counting of culturable E.coli colonies do not represent a determination of fecal microbiota.

Line 142: What do you mean by "after stewing for 3 h" for the collected blood samples?

How was the diarrhea rate determined?

Where in the intestine were the pieces for histology collected from?

How were the comparisons for E. coli made? Authors should remember bacterial counts are logaritmic and changes in log, not in gross numbers are what is meaningful.

In the discussion, authors should be cautious in expanding their findings related to E. coli to E. coli K88, since according to the methods, they used a generic approach that may be detecting E. coli different to K88.

Also, make sure results are not overstated in the discussion, since most parameters measured did not show statistical differences.

Author Response

Revision Note, (List of modification) Date: 2019-09-18 (y-m-d)

Manuscript ID: animals-595611-RI.

Dear Sir

Good day. Thank you very much for your kind consideration with our submitted article and offering us the further opportunity to submit the revised manuscript. Please find here the point to point expert reviewer’s and editor’s comments with necessary changes as per suggested with this attached file, and the amendments are highlighted in red in the revised manuscript. We have revised our manuscript for language and grammar checked by a native English speaker working in our University. We do thanks to skilled reviewers, academic editors and editorial board members as well for their critical evaluation to make the manuscript more effective for review process in Animals Journal.

Many thanks.

Sincerely yours,

Prof. Dr. Xiang Shu Piao,

State Key laboratory of Animal Nutrition, College of Animal Science and Technology, China Agricultural University, Beijing 100193, China

Corresponding Author,

Email: piaoxsh@cau.edu.cn; Tel.:+86-1062733588/Fax.: +86-1062733688

Comments: (Reviewer-2)

Comments: The study is interesting in terms of providing further evidence of the potential use of FSE in animal production. In particular, the data on nutrient digestibility is very interesting but not clearly supported by any other changes. However, there are issues with the document that need to be clarified and resolved:

Response: Thank you very much for your critical evaluation and suggestions. We have followed all of your advises in this revised submission. And we have added some possible reasons for the finding of enhanced nutrient digestibility in the discussion part.

The details are as follows:

Comments: The document needs language and grammar revision. Write in past tense when referring to experimental procedures and in present when referring to presenting results.

Response: Thank you very much. we have had the language and grammar revision. And we have followed the advices in our writing

Comments: There is no statement on intestinal morphology in the abstract, although it is in the title. Same applies for fecal microbiota.

Response: Thank you very much. I have added these parts in my abstract and title part. We added “And the villus height to crypt depth ratio in ileum was enhanced (p < 0.05) in piglets fed FSE in comparison with CTR.”

Comments: Lines 54-55. What is the evidence of residual antibiotics in pork? I believe the concerns about use of antibiotics as growth promoters is not properly addressed here.

Response: Thank you very much. I have revised this part into “However, the abuse or misuse of antibiotics in piglets’ feed can result in bacterial resistant to antibiotics and lead to the potential residues in animal products (such as pork) and the environment, which may enhance the possibility of antibiotic-resistant infections in humans.”

Comments: The pigs used in this study were weaned at 28 days, which is a bit late considering common practices and can affect the incidence of diarrhea and weight loss in weaned pigs. What was the rationale behind this decision?

Response: Thank you very much. The rationale behind this decision is rely on the situation of pig raising in China, we usually weaned at d 28.

Comments: Line 111: 12 h fasting. Why were the animals fasted before weighing?

Response: Thank you very much. To make sure every pigs at the same base when weighing and make them easier to be compared in the same base.

Comments: The terms microbiota is not used properly in the manuscript. Microbiota refers to the microbial communities found in the gastrointestinal tract. In this study, only E. coli was detected. The counting of culturable E.coli colonies do not represent a determination of fecal microbiota.

Response: Thank you very much. I may use the fecal E. coli colonies in all of this paper.

Comments: Line 142: What do you mean by "after stewing for 3 h" for the collected blood samples?

Response: Thank you very much. When we collected blood samples, we usually wait for 3 hours to let it natural settle before we centrifugate samples to better collect the serum samples which is  free of impurities.

Comments: How was the diarrhea rate determined?

Response: Thank you very much. We have added more information in this part. From d 0 to 28, the diarrhea score was monitored according to the previous described system (Pierce et al., 2005) from 1 to 5: 1, hard firm feces (rarely seen); 2, slightly soft feces; 3, soft, partially formed feces; 4, loose, semi-liquid feces (diarrhea); 5, watery, mucus-like feces (severe diarrhea). The determination of diarrhea rate was mainly depended on the average diarrhea score following the formula: Diarrhea rate (%) = diarrhea days × the number of diarrhea pigs/ (experiment days × the total number of pigs) Long et al. (2018).

Comments: Where in the intestine were the pieces for histology collected from?

Response: Thank you very much. We selected duodenum as the proximal 1/3 of the small intestine, jejunum as the 1/3 mid and ileum as 1/3 distal part.

Comments: How were the comparisons for E. coli made? Authors should remember bacterial counts are logaritmic and changes in log, not in gross numbers are what is meaningful.

Response: Thank you very much. We use the log cfu/g as unit and tested the E. coli content in fresh fecal samples was carried within a day after collection.

Comments: In the discussion, authors should be cautious in expanding their findings related to E. coli to E. coli K88, since according to the methods, they used a generic approach that may be detecting E. coli different to K88.

Response: Thank you very much. We have corrected these mistakes and be cautious in our discussion part.

Comments: Also, make sure results are not overstated in the discussion, since most parameters measured did not show statistical differences.

Response: Thank you very much. We have revised the discussion and stated the results in a more suitable way in our discussion part.

Thank you!

Round 2

Reviewer 1 Report

My comments have be addressed. I therefore suggest to publish the manuscript in its actual version.

Author Response

Revision Note, (List of modification) Date: 2019-09-20 (y-m-d)

Manuscript ID: animals-595611-R2.

Dear Sir

Good day. Thank you very much for your kind help for revising the article. We have revised our manuscript for language and grammar checked by a native English speaker working in our University. We do thanks to skilled reviewers, academic editors and editorial board members as well for their critical evaluation to make the manuscript more effective for review process in Animals Journal.

Many thanks.

Sincerely yours,

Prof. Dr. Xiang Shu Piao,

State Key laboratory of Animal Nutrition, College of Animal Science and Technology, China Agricultural University, Beijing 100193, China

Corresponding Author,

Email: piaoxsh@cau.edu.cn; Tel.:+86-1062733588/Fax.: +86-1062733688

Reviewer 2 Report

The authors address mostly the concerns indicated in my previous revision.

There are a few details to be corrected before proceeding:

A deeper English revision will improve the manuscript. There are still some wording and sentence structure issues that affect the flow. On table 3, "Apparent" should not be capitalized On Fig. 1, correct the X axis label, seems to be marking only the middle bar Table 4 and conclusions on bacterial count. I still disagree with the approach and interpretation. Considering CFU's numbers only, the differences may not represent a real reduction in E. coli, there is not even one log drop in counts.  Line 241: there is no evidence in the literature that a larger villus:crypt ratio actually indicates "better function". I suggest the authors to reconsider the word "enhanced", that has an implication of improvement, and use a word that better reflects the observation of a numerical change. Line 248 is hard to read Line 267: I think the text should say that the reduced diarrhea could be due to a reduced load of E. coli, instead of saying that "is directly due to..." since no other diarrhea causing agents were analyzed in the study. On the methods section regarding the collection points for intestinal samples, it is now stated that duodenum was considered as the first third of the intestine, jejunum the second and ileum the last third. Since that was the consideration when collecting the samples, that should remain as is. However, for the future, authors may need to consider that this distribution is not anatomically correct. The duodenum in growing pig occupies only the proximal 20-30 cm, the ileum usually not more than 40 cm proximal to the ileocecal valve, and the majority of the small intestine is jejunum. Considering young piglets, when samples are collected away from the duodenum and ileocecal valve, proper duodenum and ileum can be missed.

Author Response

Revision Note, (List of modification) Date: 2019-09-20 (y-m-d)

Manuscript ID: animals-595611-R2.

Dear Sir

Good day. Thank you very much for your kind consideration with our submitted article and offering us the further opportunity to submit the revised manuscript. Please find here the point to point expert reviewer’s and editor’s comments with necessary changes as per suggested with this attached file, and the amendments are highlighted in red in the revised manuscript. We have revised our manuscript for language and grammar checked by a native English speaker working in our University. We do thanks to skilled reviewers, academic editors and editorial board members as well for their critical evaluation to make the manuscript more effective for review process in Animals Journal.

Many thanks.

Sincerely yours,

Prof. Dr. Xiang Shu Piao,

State Key laboratory of Animal Nutrition, College of Animal Science and Technology, China Agricultural University, Beijing 100193, China

Corresponding Author,

Email: piaoxsh@cau.edu.cn; Tel.:+86-1062733588/Fax.: +86-1062733688

Comments: (Reviewer-2)

There are a few details to be corrected before proceeding:

A deeper English revision will improve the manuscript. There are still some wording and sentence structure issues that affect the flow. On table 3, "Apparent" should not be capitalized On Fig. 1, correct the X axis label, seems to be marking only the middle bar Table 4 and conclusions on bacterial count. I still disagree with the approach and interpretation. Considering CFU's numbers only, the differences may not represent a real reduction in E. coli, there is not even one log drop in counts. Line 241: there is no evidence in the literature that a larger villus: crypt ratio actually indicates "better function". I suggest the authors to reconsider the word "enhanced", that has an implication of improvement, and use a word that better reflects the observation of a numerical change. Line 248 is hard to read Line 267: I think the text should say that the reduced diarrhea could be due to a reduced load of E. coli, instead of saying that "is directly due to..." since no other diarrhea causing agents were analyzed in the study. On the methods section regarding the collection points for intestinal samples, it is now stated that duodenum was considered as the first third of the intestine, jejunum the second and ileum the last third. Since that was the consideration when collecting the samples, that should remain as is. However, for the future, authors may need to consider that this distribution is not anatomically correct. The duodenum in growing pig occupies only the proximal 20-30 cm, the ileum usually not more than 40 cm proximal to the ileocecal valve, and the majority of the small intestine is jejunum. Considering young piglets, when samples are collected away from the duodenum and ileocecal valve, proper duodenum and ileum can be missed.

Responses: Thank you very much for your critical evaluation and suggestions. We have followed your advises in this revised submission. And we have added some possible changes in our results and discussion parts.

The details are as follows:

Comments: A deeper English revision will improve the manuscript. There are still some wording and sentence structure issues that affect the flow.

Response: Thank you very much. We have had native English teacher to revise for us. And he helps us to better use the word and make better sentences.

Comments: On table 3, "Apparent" should not be capitalized On Fig. 1, correct the X axis label, seems to be marking only the middle bar

Response: Thank you very much. We have corrected the mistake on Table 3 and we have corrected the X axis label. And the 1-14 are changed into d 1 to 14, the 15-28 into d 15 to 28.

Comments: Table 4 and conclusions on bacterial count. I still disagree with the approach and interpretation. Considering CFU's numbers only, the differences may not represent a real reduction in E. coli, there is not even one log drop in counts.

Response: Thank you very much. We mentioned only the CFU's number changes of E. coli in our result. We are thankful to receive the useful comment, it will help us be better in our further study, we shall not only concern the counts with CFU numbers, but also consider the composition or other ways to better express the change of bacteria in feces. And in the discussion, we added “In our present study, we consider CFU's numbers of Escherichia coli only, the differences may not represent a real reduction in Escherichia coli, therefore, this finding still need to be further estimated in our following studies.”

Comments: Line 241: there is no evidence in the literature that a larger villus: crypt ratio actually indicates "better function". I suggest the authors to reconsider the word "enhanced", that has an implication of improvement, and use a word that better reflects the observation of a numerical change.

Response: Thank you very much. We have changed the saying to let the sentences more suitable. We have used the “numerical higher” to replace “enhanced”, and added “the villus heights of piglets fed FSE are numerical higher (p > 0.05) at approximate 11% in duodenum and 26% in jejunum, respectively” in the result. And we also added some part in discussion, “The higher numerical changes of the villus height, crypt depth and their ratio in small intestine probably indicates FSE may do benefit to a better development of small intestine in weaned piglets”.

Comments: Line 248 is hard to read Line 267: I think the text should say that the reduced diarrhea could be due to a reduced load of E. coli, instead of saying that "is directly due to..." since no other diarrhea causing agents were analyzed in the study.

Response: Thank you very much. We have changed the part in Line 248 and Line 267 to let the sentences more suitable. Line 248 was changed into “but there is no difference of performance among dietary treatments in phase 1”. Line 267 has changed into “could possibly be”

Comments: On the methods section regarding the collection points for intestinal samples, it is now stated that duodenum was considered as the first third of the intestine, jejunum the second and ileum the last third. Since that was the consideration when collecting the samples, that should remain as is.

Response: Thank you very much. We will remind this in our following studies and better consider this in the future.

Comments: However, for the future, authors may need to consider that this distribution is not anatomically correct. The duodenum in growing pig occupies only the proximal 20-30 cm, the ileum usually not more than 40 cm proximal to the ileocecal valve, and the majority of the small intestine is jejunum. Considering young piglets, when samples are collected away from the duodenum and ileocecal valve, proper duodenum and ileum can be missed.

Response: Thank you very much. We will remind this in our following studies. In our next study, we will consider to collect more suitable and representative samples from young piglets since the majority of the small intestine is jejunum, we may also collect the different part in jejunum in the future study.

Thank you!